# Low-dimensional models of neural population activity in sensory cortical circuits

**Evan Archer[1,2], Urs Köster[3], Jonathan Pillow[4], Jakob H. Macke[1,2]**
[1]Max Planck Institute for Biological Cybernetics, Tübingen
[2]Bernstein Center for Computational Neuroscience, Tübingen
[3]Redwood Center for Theoretical Neuroscience, University of California at Berkeley
[4]Princeton Neuroscience Institute, Department of Psychology, Princeton University
`evan.archer@tuebingen.mpg.de, urs@nervanasys.com`
`pillow@princeton.edu, jakob@tuebingen.mpg.de`

## Abstract

Neural responses in visual cortex are influenced by visual stimuli and by ongoing spiking activity in local circuits. An important challenge in computational neuroscience is to develop models that can account for both of these features in large multi-neuron recordings and to reveal how stimulus representations interact with and depend on cortical dynamics. Here we introduce a statistical model of neural population activity that integrates a nonlinear receptive field model with a latent dynamical model of ongoing cortical activity. This model captures temporal dynamics and correlations due to shared stimulus drive as well as common noise. Moreover, because the nonlinear stimulus inputs are mixed by the ongoing dynamics, the model can account for a multiple idiosyncratic receptive field shapes with a small number of nonlinear inputs to a low-dimensional dynamical model. We introduce a fast estimation method using online expectation maximization with Laplace approximations, for which inference scales linearly in both population size and recording duration. We test this model to multi-channel recordings from primary visual cortex and show that it accounts for neural tuning properties as well as cross-neural correlations.

## 1 Introduction

Neurons in sensory cortices organize into highly-interconnected circuits that share common input, dynamics, and function. For example, across a cortical column, neurons may share stimulus dependence as a result of sampling the same location of visual space, having similar orientation preference [1] or receptive fields with shared sub-units [2]. As a result, a substantial fraction of stimulus-information can be redundant across neurons [3]. Recent advances in electrophysiology and functional imaging allow us to simultaneously probe the responses of the neurons in a column. However, the high dimensionality and (relatively) short duration of the resulting data renders analysis a difficult statistical problem.

Recent approaches to modeling neural activity in visual cortex have focused on characterizing the responses of individual neurons by linearly projecting the stimulus on a small feature subspace that optimally drives the cell [4, 5]. Such "systems-identification" approaches seek to describe the stimulus-selectivity of single neurons separately, treating each neuron as an independent computational unit. Other studies have focused on providing probabilistic models of the dynamics of neural populations, seeking to elucidate the internal dynamics underlying neural responses [6, 7, 8, 9, 10, 11]. These approaches, however, typically do not model the effect of the stimulus (or do so using only a linear stimulus drive). To realize the potential of modern recording technologies and to progress our un-

derstanding of neural population coding, we need methods for extracting both the features that drive a neural population and the resulting population dynamics [12].

We propose the Quadratic Input Latent Dynamical System (QLDS) model, a statistical model that combines a low-dimensional representation of population dynamics [9] with a low-dimensional description of stimulus selectivity [13]. A low-dimensional dynamical system governs the population response, and receives a nonlinear (quadratic) stimulus-dependent input. We model neural spike responses as Poisson (conditional on the latent state), with exponential firing rate-nonlinearities. As a result, population dynamics and stimulus drive interact multiplicatively to modulate neural firing. By modeling dynamics and stimulus dependence, our method captures correlations in response variability while also uncovering stimulus selectivity shared across a population.

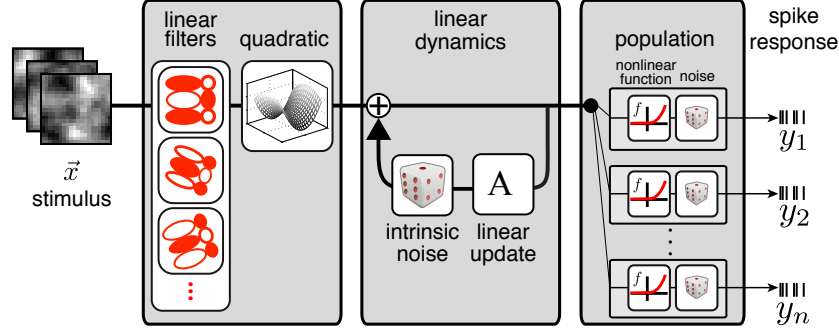

Figure 1: **Schematic illustrating the Quadratic input latent dynamical system model (QLDS).** The sensory stimulus is filtered by multiple units with quadratic stimulus selectivity (only one of which is shown) which model the feed-forward input into the population. This stimulus-drive provides input into a multi-dimensional linear dynamical system model which models recurrent dynamics and shared noise within the population. Finally, each neuron $\mathbf{y}_i$ in the population is influenced by the dynamical system via a linear readout. QLDS therefore models both the stimulus selectivity as well as the spatio-temporal correlations of the population.

## 2 The Quadratic Input Latent Dynamical System (QLDS) model

### 2.1 Model

We summarize the collective dynamics of a population using a linear, low-dimensional dynamical system with an $n$-dimensional latent state $\mathbf{x}_t$. The evolution of $\mathbf{x}_t$ is given by

$$\mathbf{x}_t = \mathbf{A}\mathbf{x}_{t-1} + f_\phi(\mathbf{h}_t) + \boldsymbol{\epsilon}_t, \tag{1}$$

where $\mathbf{A}$ is the $n \times n$ dynamics matrix and $\boldsymbol{\epsilon}$ is Gaussian innovation noise with covariance matrix $\mathbf{Q}$, $\boldsymbol{\epsilon}_t \sim \mathcal{N}(0, \mathbf{Q})$. Each stimulus $\mathbf{h}_t$ drives some dimensions of $\mathbf{x}_t$ via a nonlinear function of the stimulus, $f_\phi$, with parameters $\phi$, where the exact form of $f(\cdot)$ will be discussed below. The log firing rates $\mathbf{z}_t$ of the population couple to the latent state $\mathbf{x}_t$ via a loading matrix $\mathbf{C}$,

$$\mathbf{z}_t = \mathbf{C}\mathbf{x}_t + \mathbf{D} * \mathbf{s}_t + \mathbf{d}. \tag{2}$$

Here, we also include a second external input $\mathbf{s}_t$, which is used to model the dependence of the firing rate of each neuron on its own spiking history [14]. We define $\mathbf{D} * \mathbf{s}_t$ to be that vector whose k-th element is given by $(\mathbf{D} * \mathbf{s}_t)_k \equiv \sum_{i=1}^{N_s} \mathbf{D}_{k,i} s_{k,t-i}$. $\mathbf{D}$ therefore models single-neuron properties that are not explained by shared population dynamics, and captures neural properties such as burstiness or refractory periods. The vector $\mathbf{d}$ represents a constant, private spike rate for each neuron. The vector $\mathbf{x}_t$ represents the $n$-dimensional state of $m$ neurons. Typically $n < m$, so the model parameterizes a low-dimensional dynamics for the population.

We assume that, conditional on $\mathbf{z}_t$, the observed activity $\mathbf{y}_t$ of $m$ neurons is Poisson-distributed,

$$y_{k,t} \sim \text{Poisson}(\exp(z_{k,t})). \tag{3}$$

While the Poisson likelihood provides a realistic probabilistic model for the discrete nature of spiking responses, it makes learning and inference more challenging than it would be for a Gaussian model. As we discuss in the subsequent section, we rely on computationally-efficient approximations to perform inference under the Poisson observation model for QLDS.

## 2.2 Nonlinear stimulus dependence

Individual neurons in visual cortex respond selectively to only a small subset of stimulus features [4, 15]. Certain subpopulations of neurons, such as in a cortical column, share substantial receptive field overlap. We model such a neural subpopulation as sensitive to stimulus variation in a linear subspace of stimulus space, and seek to characterize this subspace by learning a set of basis vectors, or *receptive fields*, $\mathbf{w}_i$. In QLDS, a subset of latent states receives a nonlinear stimulus drive, each of which reflects modulation by a particular receptive field $\mathbf{w}_i$. We consider three different forms of stimulus model: a fully linear model, and two distinct quadratic models. Although it is possible to incorporate more complicated stimulus models within the QLDS framework, the quadratic models' compact parameterization and analytic elegance make them both flexible and computationally tractable. What's more, quadratic stimulus models appear in many classical models of neural computation, e.g. the Adelson-Bergen model for motion-selectivity [16]; quadratic models are also sometimes used in the classification of simple and complex cells in area V1 [4].

We express our stimulus model by the function $f_\phi(\mathbf{h}_t)$, where $\phi$ represents the set of parameters describing the stimulus filters $\mathbf{w}_i$ and mixing parameters $a_i$, $b_i$ and $c_i$ (in the case of the quadratic models). When $f_{\mathbf{B}}(\mathbf{h}_t)$ is identically 0 (no stimulus input), the QLDS with Poisson observations reduces to what has been previously studied as the Poisson Latent Dynamical System (PLDS) [17, 18, 9]. We briefly review three stimulus models we consider, and discuss their computational properties.

**Linear:** The simplest stimulus model we consider is a linear function of the stimulus,

$$f(\mathbf{h}_t) = \mathbf{B}\mathbf{h}_t, \tag{4}$$

where the rows of $\mathbf{B}$ as linear filters, and $\phi = \{\mathbf{B}\}$. This baseline model is identical to [18, 9] and captures simple cell-like receptive fields since the input to latent states is linear and the observation process is generalized linear.

**Quadratic:** Under the linear model, latent dynamics receive linear input from the stimulus along a single filter dimension, $\mathbf{w}_i$. In the quadratic model, we permit the input to each state to be a quadratic function of $\mathbf{w}_i$. We describe the quadratic by including three additional parameters per latent dimension, so that the stimulus drive takes the form

$$f_{\mathbf{B},i}(\mathbf{h}_t) = a_i \left(\mathbf{w}_i^{\mathrm{T}}\mathbf{h}_t\right)^2 + b_i \left(\mathbf{w}_i^{\mathrm{T}}\mathbf{h}_t\right) + c_i. \tag{5}$$

Here, the parameters $\phi = \{\mathbf{w}_i, a_i, b_i, c_i : i = 1, \ldots, m\}$ include multiple stimulus filters $\mathbf{w}_i$ and quadratic parameters $(a_i, b_i, c_i)$. Equation 5 might result in a stimulus input that has non-zero mean with respect to the distribution of the stimulus $\mathbf{h}_t$, which may be undesirable. Given the covariance of $\mathbf{h}_t$, it is straightforward to constrain the input to be zero-mean by setting $c_i = -a_i\mathbf{w}_i^{\mathrm{T}}\Sigma\mathbf{w}_i$, where $\Sigma$ is the covariance of $\mathbf{h}_t$ and we assume the stimulus to have zero mean as well. The quadratic model enables QLDS to capture phase-invariant responses, like those of complex cells in area V1.

**Quadratic with multiplicative interactions:** In the above model, there are no interactions between different stimulus filters, which makes it difficult to model suppressive or facilitating interactions between features [4]. Although contributions from different filters combine in the dynamics of $\mathbf{x}$, any interactions are linear. Our third stimulus model allows for multiplicative interactions between $r < m$ stimulus filters, with the $i$-th dimension of the input given by

$$f_{\phi,i}(\mathbf{h}_t) = \sum_{j=1}^{r} a_{i,j} \left(\mathbf{w}_i^{\mathrm{T}}\mathbf{h}_t\right)\left(\mathbf{w}_j^{\mathrm{T}}\mathbf{h}_t\right) + b_i \left(\mathbf{w}_i^{\mathrm{T}}\mathbf{h}_t\right) + c_i.$$

Again, we constrain this function to have zero mean by setting $c_i = -\sum_{j=1}^{r} a_{i,j} \left(\mathbf{w}_i^{\mathrm{T}}\Sigma\mathbf{w}_j\right)$.

## 2.3 Learning & Inference

We learn all parameters via the expectation-maximization (EM) algorithm. EM proceeds by alternating between expectation (E) and maximization (M) steps, iteratively maximizing a lower-bound to the log likelihood [19]. In the E-step, one infers the distribution over trajectories $\mathbf{x}_t$, given data and the parameter estimates from the previous iteration. In the M-step, one updates the current parameter estimates by maximizing the expectation of the log likelihood, a lower bound on the log likelihood. EM is a standard method for fitting latent dynamical models; however, the Poisson observation model complicates computation and requires the use of approximations.

**E-step:** With Gaussian latent states $\mathbf{x}_t$, posterior inference amounts to computing the posterior means $\boldsymbol{\mu}_t$ and covariances $\mathbf{Q}_t$ of the latent states, given data and current parameters. With Poisson observations exact inference becomes intractable, so that approximate inference has to be used [18, 20, 21, 22]. Here, we apply a global Laplace approximation [20, 9] to efficiently (linearly in experiment duration $T$) approximate the posterior distribution by a Gaussian. We note that each $f_B(\mathbf{h}_t)$ in the E-step is deterministic, making posterior inference identical to standard PLDS models. We found a small number of iterations of Newton's method sufficient to perform the E-step.

**M-step:** In the M-step, each parameter is updated using the means $\boldsymbol{\mu}_t$ and covariances $\mathbf{Q}_t$ inferred in the E-step. Given $\boldsymbol{\mu}_t$ and $\mathbf{Q}_t$, the parameters $\mathbf{A}$ and $\mathbf{Q}$ have closed-form update rules that are derived in standard texts [23]. For the Poisson likelihood, the M-step requires nonlinear optimization to update the parameters $\mathbf{C}$, $\mathbf{D}$ and $\mathbf{d}$ [18, 9]. While for linear stimulus functions $f_\phi(\mathbf{h}_t)$ the M-step has a closed-form solution, for nonlinear stimulus functions we optimize $\phi$ numerically. The objective function for $\phi$ given by

$$g(\phi) = -\frac{1}{2}\sum_{t=2}^{T}\left[(\boldsymbol{\mu}_t - \mathbf{A}\boldsymbol{\mu}_{t-1} - f_\phi(\mathbf{h}_t))^{\mathrm{T}}\mathbf{Q}^{-1}(\boldsymbol{\mu}_t - \mathbf{A}\boldsymbol{\mu}_{t-1} - f_\phi(\mathbf{h}_t))\right] + \mathrm{const.},$$

where $\boldsymbol{\mu}_t = \mathbb{E}[\mathbf{x}_t|\mathbf{y}_{t-1}, \mathbf{h}_t]$. If $\phi$ is represented as a vector concatenating all of its parameters, the gradient of $g(\phi)$ takes the form

$$\frac{\partial g(\phi)}{\partial \phi} = -\mathbf{Q}^{-1}\sum_{t=2}^{T}(\boldsymbol{\mu}_t - \mathbf{A}\boldsymbol{\mu}_{t-1} - f_\phi(\mathbf{h}_t))\frac{\partial f(\mathbf{h}_t)}{\partial \phi}. \tag{6}$$

For the quadratic nonlinearity, the gradients with respect to $f(\mathbf{h}_t)$ take the form

$$\frac{\partial f(\mathbf{h}_t)}{\partial \mathbf{w}_i} = 2\left[a_i\left(\mathbf{h}_t{}^{\mathrm{T}}\mathbf{w}_i\right) + b_i\right]\mathbf{h}_t{}^{\mathrm{T}}, \qquad \frac{\partial f(\mathbf{h}_t)}{\partial a_i} = \left(\mathbf{h}_t{}^{\mathrm{T}}\mathbf{w}_i\right)^2, \tag{7}$$

$$\frac{\partial f(\mathbf{h}_t)}{\partial b_i} = \mathbf{h}_t{}^{\mathrm{T}}\mathbf{w}_i, \qquad \frac{\partial f(\mathbf{h}_t)}{\partial c_i} = 1. \tag{8}$$

Gradients for the quadratic model with multiplicative interactions take a similar form. When constrained to be 0-mean, the gradient for $c_i$ disappears, and is replaced by an additional term in the gradients for $\mathbf{a}$ and $\mathbf{w}_i$ (arising from the constraint on $\mathbf{c}$).

We found both computation time and quality of fit for QLDS to depend strongly upon the optimization procedure used. For long time series, we split the data into small minibatches. The QLDS E-step and M-step each naturally parallelize across minibatches. Neurophysiological experiments are often naturally segmented into separate trials across different stimuli and experimental conditions, making it possible to select minibatches without boundary effects.

## 3 Application to simulated data

We illustrate the properties of QLDS using a simulated population recording of 100 neurons, each responding to a visual stimulus of binary, white spatio-temporal noise of dimensionality 240. We simulated a recording with $T = 50000$ samples and a 10-dimensional latent dynamical state. Five of the latent states received stimulus input from a bank of 5 stimulus filters (see Fig. 2**A**, top row), and the remaining latent dimensions only had recurrent dynamics and noise. We aimed to approximate the properties of real neural populations in early sensory cortex. In particular, we set the dynamics matrix $\mathbf{A}$ by fitting the model to a single neuron recording from V1 [4]. When fitting the model, we assumed the same dimensionalities (10 latent states, 5 stimulus inputs) as those used to generate the data. We ran 100 iterations of EM, which—-for the recording length and dimensionality of this system—took about an hour on a 12–core intel Xeon CPU at 3.5GHz.

The model recovered by EM matched the statistics of the true model well. Linear dynamical system and quadratic models of stimulus selectivity both commonly have invariances that render a particular parameterization unidentifiable [4, 15], and QLDS is no exception: the latent state (and its parameters) can be rotated without changing the model's properties. Hence it is possible only to compare the *subspace* recovered by the model, and not the individual filters. In order to visualize subspace recovery, we computed the best $\ell_2$ approximation of the 5 "true" filters in the subspace spanned by

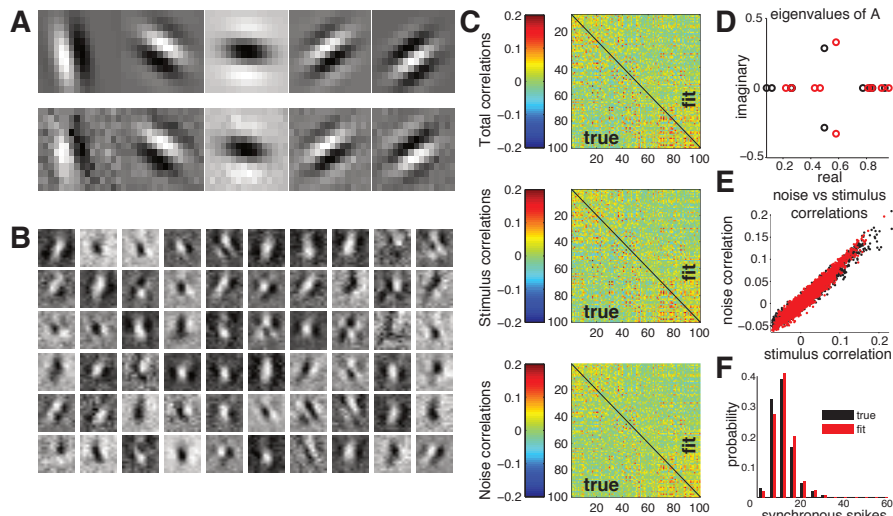

Figure 2: **Results on simulated data.** Low-dimensional subspace recovery from a population of 100 simulated neurons in response to a white noise stimulus. **(A)** Simulated neurons receive shared input from 5 spatio-temporal receptive fields (top row). QLDS recovers a subspace capable of representing the original 5 filters (bottom row). **(B)** QLDS permits a more compact representation than the conventional approach of mapping receptive fields for each neuron. For comparison with the representation in panel A, we here show the spike-triggered averages of the first 60 neurons in the population. **(C)** QLDS also models shared variability across neurons, as visualised here by the three different measures of correlation. Top: Total correlation coefficients between each pair of neurons. Values below the diagonal are from the simulated data, above the diagonal correspond to correlations recovered by the model. Center: Stimulus correlations Bottom: Noise correlations. **(D)** Eigenvalues of dynamics matrix $A$ (black is ground truth, red is estimated). **(E)** In this model, stimulus and noise correlations are dependent on each other, for the parameters chosen in this stimulation, there is a linear relationship between them. **(F)** Distribution of population spike counts, i.e. total number of spikes in each time bin across the population.

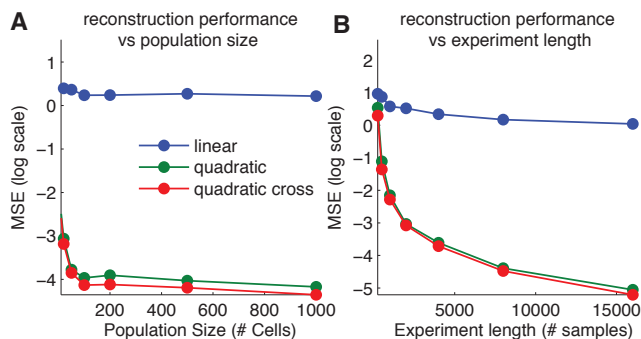

Figure 3: Recovery of stimulus subspace as a function of population size **(A)** and experiment duration **(B)**. Each point represents the best filter reconstruction performance of QLDS over 20 distinct simulations from the same "true" model, each initialized randomly and fit using the same number of EM iterations. Models were fit with each of three distinct stimulus nonlinearities, linear s (blue), quadratic (green), and quadratic with multiplicative interactions (red). Stimulus input of the "true" was a quadratic with multiplicative interactions, and therefore we expect only the multiplicative model (red) to each low error rates.

the estimated $\hat{\mathbf{w}}_i$ (see Fig. 2 **A** bottom row). In QLDS, different neurons share different filters, and therefore these 5 filters provide a compact description of the stimulus selectivity of the population [24]. In contrast, for traditional single-neuron analyses [4] 'fully-connected' models such as GLMs [14] one would estimate the receptive fields of each of the 100 filters in the population, resulting in a much less compact representation with an order of magnitude more parameters for the stimulus-part alone (see Fig. 2**B**).

QLDS captures both the stimulus-selectivity of a population and correlations across neurons. In studies of neural coding, correlations between neurons (Fig. 2**C**, top) are often divided into stimulus-correlations and noise-correlations. Stimulus correlations capture correlations explainable by similarity in stimulus dependence (and are calculated by shuffling trials), whereas noise-correlations capture correlations not explainable by shared stimulus drive (which are calculated by correlating residuals after subtracting the mean firing rate across multiple presentations of the same stimulus). The QLDS-model was able to recover both the total, stimulus and noise correlations in our simulation (Fig. 2**C**), although it was fit only to a single recording without stimulus repeats. Finally, the model also recovered the eigenvalues of the dynamics (Fig. 2**D**), the relationship between noise and stimulus correlations (Fig. 2**E**) and the distribution of population spike counts (Fig. 2**F**).

We assume that all stimulus dependence is captured by the subspace parameterized by the filters of the stimulus model. If this assumption holds, increasing the size of the population increases statistical power and makes identification of the stimulus selectivity easier rather than harder, in a manner similar to that of increasing the duration of the experiment. To illustrate this point, we generated multiple data-sets with larger population sizes, or with longer recording times, and show that both scenarios lead to improvements in subspace-recovery (see Fig. 3).

## 4    Applications to Neural Data

**Cat V1 with white noise stimulus**    We evaluate the performance of the QLDS on multi-electrode recordings from cat primary visual cortex. Data were recorded from anaesthetized cats in response to a single repeat of a 20 minute long, full-field binary noise movie, presented at 30 frames per second, and 60 repeats of a 30s long natural movie presented at 150 frames per second. Spiking activity was binned at the frame rate (33 ms for noise, 6.6 ms for natural movies). For noise, we used the first 18000 samples for training, and 5000 samples for model validation. For the natural movie, 40 repeats were used for training and 20 for validation. Silicon polytrodes (Neuronexus) were employed to record multi-unit activity (MUA) from a single cortical column, spanning all cortical layers with 32 channels. Details of the recording procedure are described elsewhere [25]. For our analyses, we used MUA without further spike-sorting from 22 channels for noise data and 25 channels for natural movies. We fit a QLDS with 3 stimulus filters, and in each case a 10-dimensional latent state, i.e. 7 of the latent dimensions received no stimulus drive.

Spike trains in this data-set exhibited "burst-like" events in which multiple units were simultaneously active (Fig. 4**A**). The model captured these events by using a dimension of the latent state with substantial innovation noise, leading substantial variability in population activity across repeated stimulus presentations. We also calculated pairwise (time-lagged) cross-correlations for each unit pair, as well as the auto-correlation function for each unit in the data (Fig. 4**B**, 7 out of 22 neurons shown, results for other units are qualitatively similar.). We found that samples from the model (Fig. 4**B**, red) closely matched the correlations of the data for most units and unit-pairs, indicating the QLDS provided an accurate representation of the spatio-temporal correlation structure of the population recording. The instantaneous correlation matrix across all 22 cells was very similar between the physiological and sampled data (Fig. 4**C**). We note that receptive fields (Fig. 4**F**) in this data did not have spatio-temporal profiles typical of neurons in cat V1 (this was also found when using conventional analyses such as spike-triggered covariance). Upon inspection, this was likely a consequence of an LGN afferent also being included in the raw MUA. In our analysis, a 3-feature model captured stimulus correlations (in held out data) more accurately than 1- and 2- filter models. However, 10-fold cross validation revealed that 2- and 3- filter models do not improve upon a 1-filter model in terms of one-step-ahead prediction performance (i.e. trying to predict neural activity on the next time-step using past observations of population activity and the stimulus).

**Macaque V1 with drifting grating stimulus:**    We wanted to evaluate the ability of the model to capture the correlation structure (i.e. noise and signal correlations) of a data-set containing multiple repetitions of each stimulus. To this end, we fit QLDS with a Poisson observation model to the population activity of 113 V1 neurons from an anaesthetized macaque, as described in [26]. Drifting grating stimuli were presented for 1280ms, followed by a 1280ms blank period, with each of 72 grating orientations repeated 50 times. We fit a QLDS with a 20-dimensional latent state and 15 stimulus filters, where the stimulus was paramterized as a set of phase-shifted sinusoids at the appropriate spatial and temporal frequency (making $\mathbf{h}_t$ 112-dimensional). We fit the QLDS to 35 repeats,

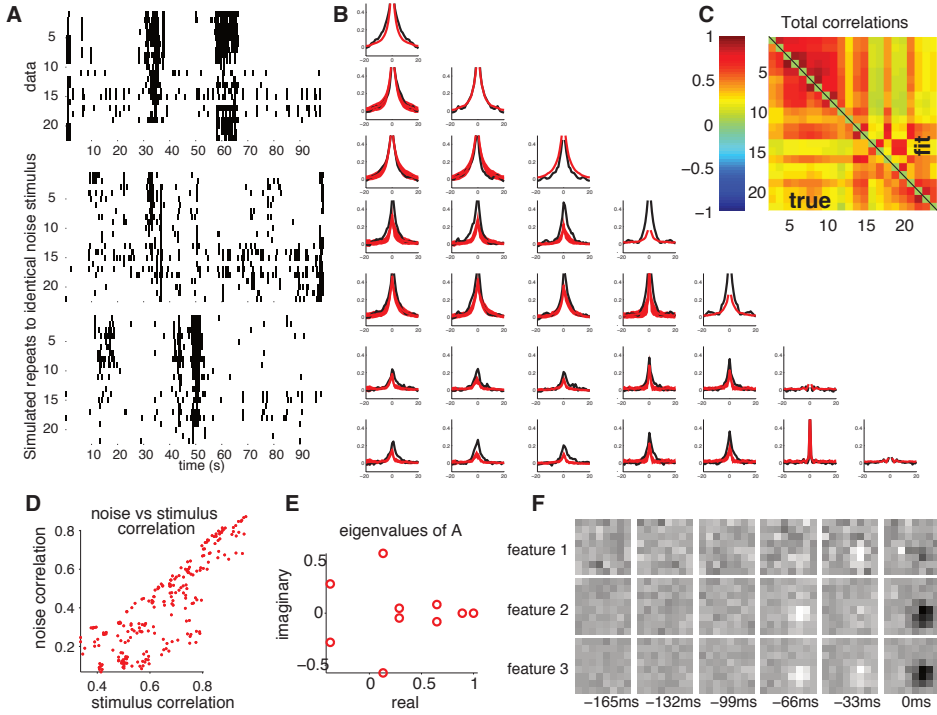

Figure 4: **QLDS fit to V1 cells with noise stimuli.** We fit QLDS to $T = 18000$ samples of 22 neurons responding to a white noise stimulus, data binned at 33 ms. We used the quadratic with multiplicative interactions as the stimulus nonlinearity. The QLDS has a 10-dimensional latent state with 3 stimulus inputs. All results shown here are compared against $T = 5000$ samples of test-data, not used to train the model. **(A)** Top row: Rasters from recordings from 22 cells in cat visual cortex, where cell index appears on the $y$ axis, and time in seconds on the $x$. Second and third row: Two independent samples from the QLDS model responding to the same noise stimuli. Note that responses are highly variable across trials. **(B)** Auto- and cross-correlations for data (black) and model (red) cells. For the model, we average across 60 independent samples, thickness of red curves reflects 1 standard deviation from the mean. Panel $(i, j)$ corresponds to cross-correlation between units with indices $i$ and $j$, panels along the diagonal show auto-correlations. **(C)** Total correlations for the true (lower diagonal) and model (upper diagonal) populations. **(D)** Noise correlations scattered against stimulus correlations for the model. As we did not have repeat data for this population, we were not able to reliably estimate noise correlations, and thereby evaluate the accuracy of this model-based prediction. **(E)** Eigenvalues of the dynamics matrix $\mathbf{A}$. **(F)** Three stimulus filters recovered by QLDS. We selected the 3-filter QLDS by inspection, having observed that fitting with larger number of stimulus filters did not improve the fit. We note that although two of the filters appear similar, that they drive separate latent dimensions with distinct mixing weights $a_i$, $b_i$ and $c_i$.

and held out 15 for validation. The QLDS accurately captured the stimulus and noise correlations of the full population (Fig. 5**A**). Further, a QLDS with 15 shared receptive fields captured simple and complex cell behavior of all 113 cells, as well as response variation across orientation (Fig. 5**B**).

## 5 Discussion

We presented QLDS, a statistical model for neural population recordings from sensory cortex that combines low-dimensional, quadratic stimulus dependence with a linear dynamical system model. The stimulus model can capture simple and complex cell responses, while the linear dynamics capture temporal dynamics of the population and shared variability between neurons. We applied QLDS to population recordings from primary visual cortex (V1). The cortical microcircuit in V1 consists of highly-interconnected cells that share receptive field properties such as orientation preference [27], with a well-studied laminar organization [1]. Layer IV cells have simple cell receptive field properties, sending excitatory connections to complex cells in the deep and superficial layers. Quadratic

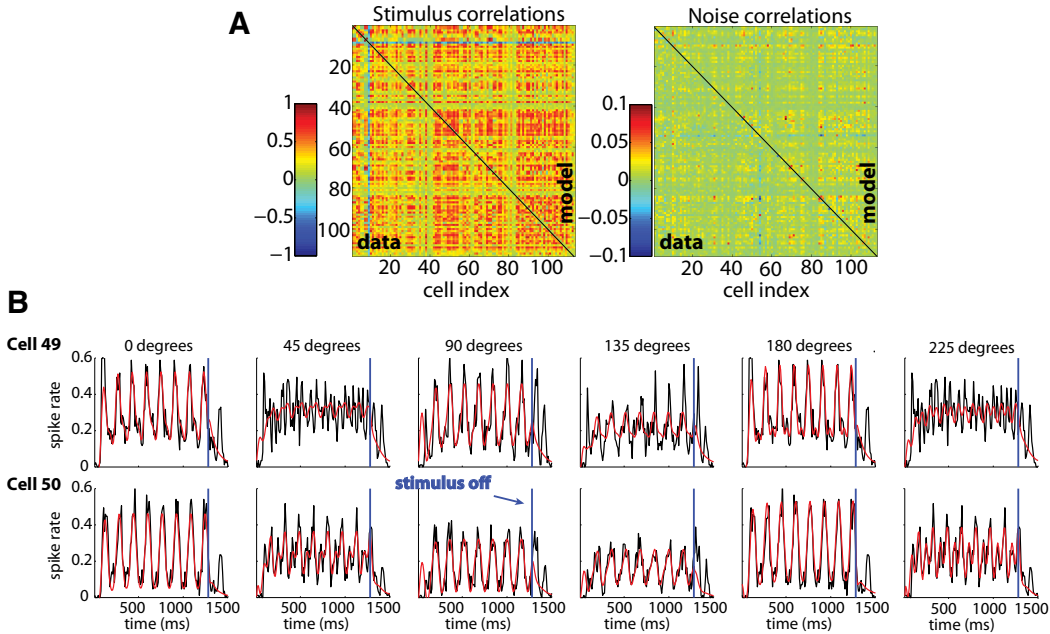

Figure 5: **QLDS fit to** 113 **V1 cells across** 35 **repeats of each of** 72 **grating orientations. (A)** Comparison of total correlations in the data and generated from the model, **(B)** For two cells (cells 49 and 50, using the index scheme from **A**) and 6 orientations (0, 45, 90, 135, 180, and 225 degrees), we show the posterior mean prediction performance (red traces) in in comparison to the average across 15 held-out trials (black traces). In each block, we show predicted and actual spike rate (y-axis) over time binned at 10 ms (x-axis). Stimulus offset is denoted by a vertical blue line.

stimulus models such as the classical "energy model" [16] of complex cells reflect this structure. The motivation of QLDS is to provide a statistical description of receptive fields in the different cortical layers, and to parsimoniously capture both stimulus dependence and correlations across an entire population.

Another prominent neural population model is the GLM (Generalized Linear Model, e.g. [14]; or the "common input model", [28]), which includes a separate receptive field for each neuron, as well as spike coupling terms between neurons. While the GLM is a successful model of a population's statistical response properties, its fully–connected parameterization scales quadratically with population size. Furthermore, the GLM supposes direct couplings between pairs of neurons, while monosynaptic couplings are statistically unlikely for recordings from a small number of neurons embedded in a large network.

In QLDS, latent dynamics mediate both stimulus and noise correlations. This reflects the structure of the cortex, where recurrent connectivity gives rise to both stimulus-dependent and independent correlations. Without modeling a separate receptive field for each neuron, the model complexity of QLDS grows only linearly in population size, rather than quadratically as in fully-connected models such as the GLM [14]. Conceptually, our modeling approach treats the entire recorded population as a single "computational unit", and aims to characterize its joint feature-selectivity and dynamics. Neurophysiology and neural coding are progressing toward recording and analyzing datasets of ever larger scale. Population-level parameterizations, such as QLDS, provide a scalable strategy for representing and analyzing the collective computational properties of neural populations.

**Acknowledgements**

We are thankful to Arnulf Graf and the co-authors of [26] for sharing the data used in Fig. 5, and to Memming Park for comments on the manuscript. JHM and EA were funded by the German Federal Ministry of Education and Research (BMBF; FKZ: 01GQ1002, Bernstein Center Tübingen) and the Max Planck Society, and UK by National Eye Institute grant #EY019965. Collaboration between EA, JP and JHM initiated at the 'MCN' Course at the Marine Biological Laboratory, Woods Hole.

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
