[Reviews · NeurIPS 2014]

Submitted by Assigned_Reviewer_27

One of the chief concerns of systems neuroscientists is characterising how individual neurons respond to sensory stimuli. Since the stimulus space is often huge, data is always limited, and neurons are fundamentally noisy, the statistical challenges involved with this characterisation have spurned a vibrant field of computational neuroscience. This paper considers a particular form of this task, where recordings are made of local populations of neurons, at some intermediate point in the processing hierarchy in the brain (e.g. neurons within a cortical column, within a primary cortical sensory area). Such recordings are very common already, and also are growing in number and fidelity. The paper argues that under these conditions, an assumption can often be made that the neurons under study are receiving inputs from a smaller number of common channels, which are both stimulus-driven and subject to recurrent dynamics. The authors proceed to formalise this as a statistically-reasoned generative model for the observed neural responses (as spike counts), and present a Laplace/EM inference scheme to invert it given population data. The authors validate the inference on a simulated data set, and then fit it to V1 data.

Overall, I think this is a well-motivated problem, and the solution the authors offer is both novel and interesting. The paper is clear and well-written, and makes a useful contribution to the field. There are a few issues here and there, which I list below.

More importantly, I think there's some weakness in the scientific interpretations of these results that the authors could clean up. The construction the paper champions is essentially a beefed-up subunit model. Subunit models have a long and rich history in systems neuroscience, starting at least as early as Barlow & Levick (1964) or Hochstein & Shapley (1976) (perhaps we should include Hubel & Wiesel 1959). These are also having renewed interest as hierarchical models of feedforward processing, e.g. Mineault et al (2012). The first stage of the proposed model in Figure 1 is essentially a collection of nonlinear subunits. The authors do flag subunits in the second sentence of the introduction, but this link (and terminology) is dropped beyond this point. Now, of course, the beef this paper adds is important and useful, and I am a fan. However, I think that the systems neuroscience community would be more willing to engage with this work and build upon it if the subunit idea were carried through, especially because it captures an intuition about the feedforward processing circuits in the brain. And, in the absence of an argument to the contrary, I think it would be worthwhile to use the terminology of subunits to refer to this first stage within the model. Some of these subunits, of course, would not be carrying stimulus-driven information, only latent dynamical signals; this shouldn't be a big deal.

Given that, I think there is some interpretation that needs to be made of the results of the simulation (Fig 2A-B especially), and the real data (Fig 4F especially). In the simulation, the ground truth (and recovered) subunits look like simple cells; in the real recordings, the recovered subunit space looks more like LGN inputs (albeit without a visible surround). As much as I rotate these latter recovered features in my head to get around the identifiability issue, I can't make the two examples match. By the time Section 4 has come around, the authors have set everything in place for a big reveal of simple-cell subunits, and it's rather disappointing that the results don't deliver. Moreover, it's not clear how the recovered features would generate the type and diversity of V1 receptive fields we're expecting. There are many potential reasons why this might be the case (e.g. sampling biases; all the units were simple), but the authors are in a better position than I to provide a reasonable explanation of this. To make things stranger, the first paragraph of the discussion seems to claim that the model can capture both simple cells and complex cells, and that it does. This was not the impression I came away with from Figure 4. Ultimately, perhaps the paper is principly about a model, but V1 is probably the most well-understood model testbed for neural systems identification, and there are expectations for the results here. If no explanation is given to intuitively validate the recovered features, readers will conclude that it doesn't recover what's expected here, so it won't work elsewhere.

Quality: The paper is technically sound, with a couple of minor typographical errors listed below.

Aside from the larger issue above, one lingering question I had was how to choose the number of dimensions in the first layer. The analysis in Fig. 3 (and surrounding text) seems like the right place to do this, but as far as I can tell, the number of stimulus-driven and stimulus-independent subunits were fixed at the ground truth. There should be some discussion of what happens when this is underestimated or overestimated. Figure 3 does not provide good enough evidence, as this assumes the ground truth (and the trends are hard to see in 3A).

Minor errors:
- eq. (2) and line 100-2: should $\vec{s}_t$ be the spike count vector $\vec{y}_t$, for consistency with eq. (3), and the typically implementation of the spike feedback kernel? Then this vector should also be integer-valued not real-valued.
- line 183: C/D/d should be boldfaced (as matrix/matrix/vector), not scalar
- eqn (7), LHS: the $\vec{w}_t^T$ should be a $\vec{h}_t^T$. Still, I think these equations could be omitted since the derivatives are pretty simple.

Clarity: In general, the paper is written clearly and organised well. There were a few points throughout the text where I felt some extra clarification would make it read better:

- eqn (2) and below: $\vec{D} \cdot \vec{s}_t$ -- I would suggest using a convolution symbol $\ast$ in place of $\cdot$ to convey its intuition. Also the constant $\vec{d}$ is not annotated.
- line 115-117: There's some ambiguity about what refers to individual neurons, what refers to the local (recorded/columnar) subpopulation, and what refers to all of V1. There's at least one way of misinterpreting what is being said here. Individual neurons may care only about a stimulus subspace (hence STC). A local subpopulation will care about a larger stimulus subpsace than any one neuron (though not proportionally larger). The whole of V1 probably cares about a substantial portion of the whole stimulus space. The important aspect of *this* model is that it learns a basis for the local subpopulation's responses, which is assumed to be reasonably small. This is captured in the introduction to the paper, but is obscured here.
- line 126/127, and subsequent: I found it confusing to talk about $\vec{B}$ as the concatenation of all /nonlinearity/ parameters, when $f$ could be a linear function. This is an instance where /subunit/ parameters would work better.
- line 146: make this "where the parameters B = {...} includes"
- line 166: "one infers the distribution of trajectories," -- insert $\vec{x}_t$ here. Likewise, start of line 168, "estimates" -- insert a $\vec{B}$ here.
- line 172/173: the covariances $\vec{Q}_t$ of the latent states. Better to call this $\vec{\Lambda}_t$ or $\vec{\Sigma}_t$ or something like that, to avoid conflation with the covariance of epsilon_t. Line 181 is currently particularly confusing with both Q_t and Q.
- further to that, I wasn't entirely clear what the posterior covariances $\vec{Q}_t$ should look like. This seems like it could be of some importance with the minibatch procedure. Also is it correct that the posterior covariance on the latent states only affects inference on the dynamical system parameters, and not on the subunit/nonlinearity parameters? Beyond this, I didn't feel like I got a good sense of how the information was combined across minibatches to update both sets of parameters in the M-step.
- Fig 2C: wasn't clear exactly how the noise correlations were estimated without repeats (as for 4D).
- Fig 2E: some info is necessary on how to interpret this plot, and what its relevance is. Same again for 4D and 5D. This appears to be playing the role of a validating result, but for Fig 4D and 5D I have no idea why this conveys as much.
- line 272: I was amused by the precision in the computation time, but I wasn't sure how to extrapolate from this anecdote.

Originality: tick, but some work needed to link to subunit literature (see above).

Significance: tick, but some work needed to link to subunit literature (see above).
Summary: The paper presents an interesting model that is technically sound and works well. Some changes should be made to give more scientific insight, especially given that it is hard to interpret the results on the real data.

Submitted by Assigned_Reviewer_35

This paper presents a novel modeling framework that combines two research directions: low-dimensional models of ongoing cortical activity and finding relevant stimulus features for sensory neurons. The idea is potentially very useful, but serious concerns about the validity of the presented approach exist.

1) The authors state that increasing the size of the recurrent population increases statistical power and makes identification of the stimulus selectivity easier rather than harder. First, this statement is not supported by the simulations presented in Figure 3A (except in the case of the linear model, and even in that case the effect is weak). Second, and most importantly, this statement cannot be true conceptually if the number of independent receptive fields increases with the size the recurrent population, which seems like a plausible scenario for the primary visual cortex.

2) The choice of the number of relevant features that drive the recurrent network seems to be made arbitrarily. For example, during the analysis of V1 noise data, three stimulus features are selected. Yet, the analysis reveals that only two features are linearly independent and the third feature is barely visible above the noise (Figure 4F). If taken at face value, the analysis would imply that the 22 V1 neurons can be characterized by one receptive field.

3) The are substantial deviations between the experiments and modeling in terms of stimulus correlations (Fig. 5B).
Summary: An interesting approach but the number of features seems to be selected arbitrarily. How reconstruction might be affected by the presumed number of features was not discussed.

Submitted by Assigned_Reviewer_36

The authors developed a method of accounting for the transformation of visual stimulus to spiking activity of a large population of neurons using a latent dynamical system model of ongoing cortical activity developed in refs [7-12]. The point of this contribution would be to introduce a nonlinear stimulus dependence called the quadratic latent dynamical system (QLDS) in Eq.(5). Accordingly the performance was significantly improved as demonstrated in Fig.3. It appears that the estimation is performed efficiently with the online expectation maximization with Laplace approximations. The method was applied to the multi-channel data in the primary visual cortex. The developed method and analysis appear solid and the results are fine. I think the paper deserves publication.
Summary: A good contribution. A nonlinear extension of stimulus dependence called the quadratic latent dynamical system (QLDS). The significant improvement is achieved by the proposed method.
Author Feedback
Author rebuttal: We thank the reviewers for careful and constructive evaluations, which we believe will significantly improve the manuscript. We are grateful to Reviewer 36 for the positive assessment, and focus on the comments of Reviewers 27 and 35, beginning with shared concerns:

1) How many features?
We apologize for being unclear about how to determine the dimensionality of the feature space in QLDS. We used cross-validation (CV) to select the number of features both in simulation and in application to data. Briefly, we found CV to work well when selecting model complexity for simulated data. Additionally, when we apply the model to a population of retinal ganglion cells tiling visual space, we do not find the retina to be low-dimensional: as many features are required as there are cells. Further, we find QLDS to be robust to model misspecification, in that performance decreased only modestly if the dimensionality was too high (results not shown in draft).

2) Fig. 3A, performance as a function of population size:
We apologize for the lack of clarity in Fig. 3. While Fig 3A shows rapid early performance improvements as a function of population size, local minima issues cause the “bumpy” non-monotonic behavior at large population sizes. We re-generated the figure with a more extensive simulation, and after eliminating some issues with local minima which had caused some of the high-N fits to be suboptimal. As expected, the figure then shows an increase in performance with population size for fixed dimensionality of the stimulus drive.

3) Surprising low-dimensionality and receptive field in empirical data:
We thank reviewers 27 and 35 for highlighting a puzzling feature of the neural data set we analyzed. In this population recording (from a vertical probe in cat V1), the recovered filters showed ‘LGN like’ filters but not the expected “V1-like” filters. Additional analyses, in consultation with our experimental collaborator, have revealed that this is an outlier-- indeed, this particular property of the data also appears in single-neuron analyses such as spike-triggered average or covariance (and might reflect contamination from an LGN afferent, explaining both that one filter is sufficient and that the filter does not look ‘V1-like’). Thus, we think it is a problem with the particular data-set, not with our method. On other experimental sessions from the same set of experiments, we find richer feature selectivity and higher dimensionality, both with conventional methods and QLDS. We are confident that a revised version of Fig. 4 will compellingly illustrate our points.

Reviewer 27:

We thank the reviewer for the particularly thoughtful review and detailed suggestions.

a) We agree that our work is a subunit model (as motivated and referenced), but that we do need to make more explicit the connections with this body of work. We also like the suggestion to explicitly adopt the subunit terminology (though care needs to be taken in defining subunit, given latent variable models’ identifiability issues).

b) Receptive fields: The simulation was designed to have receptive field filters which are visually like those in (Rust et al, Neuron 2005). Regarding the strange-looking receptive fields in the dataset used for Fig. 4, please see our comment above.

c) QLDS can indeed capture the responses of both simple and complex cells. In addition to our simulation results, we have validated this point by fitting the QLDS to individual complex cells and comparing it to a method for predicting the single-neuron responses to visual input, the Generalized Quadratic Model (GQM) (Park et al, 2013). For both QLDS and GQM, we quantified how well neural responses can be predicted using the stimulus and recent history of spiking (which, for GQM, was incorporated using history features). We found that QLDS performs as well as GQM when modeling the responses of simple and complex cells.

Reviewer 35:

a) We apologize for failing to describe how we selected the number of features for QLDS. As we describe above, we found standard CV to work well, and QLDS to be robust to the dimensionality being set too large.

Fig. 4 shows a 3-feature model, which we found to capture stimulus correlations, in held out data, more effectively than 1- and 2- filter models. However, the reviewer is right to suggest that we select the number of features automatically, and 10-fold CV indeed reveals a single filter to be sufficient for one-step-ahead prediction (though, as pointed out above, this is a feature of this particular dataset). In retrospect, we agree with the reviewer that a single filter would have been more intuitive to show; however, we stress that the predictive performance of the model is not highly dependant upon the number of features.

b) The reviewer is correct that, under the assumption of linearly independent filters across a population, statistical power does not increase with population size. We do not expect this to be the case for the entirety of V1, but we do expect it to be the case here. Critically, our aim is to model a vertically-aligned V1 subpopulation; neurons within the same column have receptive field overlap, the same orientation tuning, and shared subunits. Hence, these cells do have shared stimulus selectivity, which gives rise to redundancy in responses. In application to cortical data, we find that a relatively (to population size) small number of filters are sufficient for describing the selectivity of a population.

c) Fig. 5B: QLDS captures both total and noise correlations in responses to natural movies (challenging stimuli for any neural system identification approach), and also the qualitative relationship between stimulus and noise correlations on this data. However, there are notable deviations in the estimated and true stimulus correlations. Based on the suggestion of the reviewer, we ran this analysis again and for more iterations; this visibly improves the fit.